# Assessing the asymmetric war-growth nexus: A case of Afghanistan

**Mohammad Ajmal Hameed**⦿*, **Mohammad Mafizur Rahman, Rasheda Khanam**

School of Business, University of Southern Queensland, Toowoomba, Queensland, Australia

* U1129358@usq.edu.au

**Data Availability Statement:** Our dataset is publicly published in the Mendeley repository and can be found via the following link: https://data.mendeley.com/datasets/2hthy8r3dt/1 or the following DOI: DOI:10.17632/2hthy8r3dt.1.

## Abstract

This study explores the war-growth nexus in Afghanistan, a country where war-torn acts inform resource allocation. Employing the asymmetric ARDL, dynamic multipliers, and asymmetric causality techniques, the initial results confirm the existence of a long-run asymmetric nexus amid predictors. The asymmetric ARDL results indicate that a positive asymmetric shock from the per capita cost of war reduces per capita GDP—that is, economic growth—while a negative asymmetric shock from the per capita cost of war increases growth in the short and long run. Moreover, the findings reveal that per capita capital investment, per capita energy consumption, per capita household consumption, per capita remittance, per capita foreign direct investment, population growth, and inflation rate have significantly asymmetric effects on growth, highlighting non-monotonic impacts in scale and magnitude. The results of the asymmetric causality technique by bootstrap confirm that there is an asymmetric bidirectional causality between growth, per capita cost of war, per capita household consumption, per capita capital investment, and per capita foreign direct investment, while expanding only unidirectional causality with per capita remittance, population growth, and inflation rate. Based on the findings, the study concludes by offering relevant policy recommendations.

## 1 Introduction

Assessing the impact of war on the macroeconomic variables of a nation that has experienced a long-run armed conflict is a critical research topic that has recently gained popularity among academics, scholars, and policymakers. Undoubtedly, many nations have witnessed civil wars that resulted in the destruction of economies, infrastructures, and social development backwardness. Though understanding the effects of civil war is central to the developmental policies of post-conflict environments, little is known about their scale and magnitude [1]. The negative impacts of war occur either concurrently during the period of war or develop as enduring effects after it ends—thereby hampering the economy over the long term. On the one hand, statistical evidence shows that as a result of civil war, budgetary reallocation from physical and technological development to military operations swiftly impacts the economy in the short-run [2], while on the other hand, the destruction of infrastructure and human capital flight are the enduring negative impacts of war on an economy. In the context of Afghanistan, rising military spending is an opportunity cost, diverting funds that could otherwise be spent

**Funding:** The author(s) received no specific funding for this work.

**Competing interests:** The authors have declared that no competing interests exist.

**Abbreviations:** ADF, Augmented Dickey-Fuller; AIC, Akaike information criterion; ARDL, Autoregressive distributed lags; GDP, Gross domestic product; GFCE, Per capita final household consumption expenditure; HQIC, Hannan-Quinn information criterion; INR, Inflation rate; NARDL, Non-linear autoregressive distributed lags; PCCE, Per capita capital investment; PCEC, Per capita energy consumption; PCOW, Per capita cost of war; PFDI, Per capita FDI; PGDP, Per capita gross domestic product; PGR, Population growth rate; PP, Phillips-Perron; QLD, Queensland; SIC, Schwarz information criterion; WB, World Bank; WDI, World development indicators.

on essential public services, such as state-building, education, health care, private sector development and promotion, and the development of infrastructure for its citizens and businesses to ease economic growth. Because fewer fiscal revenues limit the Afghan government's ability to offer these fundamental public services to its citizens, financial losses add to the conflicts of humanitarian consequences. This loss appeared to be significant in Afghanistan and resulted in greater economic depression and a steady state failure [3]. According to the International Monetary Fund [4], the long-run war in Afghanistan lowered yearly national income by roughly 50% in 2016. This amounts to nearly 70 billion Afghanis (1 billion US dollars) and is based on the level of violence in 2005.

Moreover, in a war-state economy, governments are hesitant to restrict monetary policies such as tax rises or subsidies on commodities since such regulations will indeed bring home the instantaneous wars; as a result of the government's unwillingness or inability to interrogate its politics straight with current war costs, price pressures are likely to increase. Djankov and Reynal-Querol [5] argue that poverty has a causal relationship with public support for insurgent groups and therefore with the feasibility of this form of spending. Most aid spending by governments aiming to restore the balance of power is predicated on the opportunity-cost notion of diverting prospective employees, they claim. The theory goes that unemployed young people are much less likely to engage in political violence, establishing a link between unemployment and war in areas where insurgent groups are active. Therefore, its continuity entails several negative consequences for the overall economic performance in several ways. First, in most respects, the government declines spending in all sectors except military operations. Second, household entitlements decline, resulting in a downward shift both in aggregate demand and aggregate supply. Third, the economy conforms to a war-torn society [6]. From an economic theory point of view, there is no general consensus on the duration of the effects of war on the economy. The neoclassical growth theory—factors of production—says an economy swiftly recovers and returns to its steady state after the war ends. Alternative models suggest that the return takes a long time because of the slow recovery of the economy [7] and [8], or that countries can be stuck in a low-level equilibrium where war and loss of production factors coexist [9]. At minimum, two key factors may establish this contradiction. First, the duration of wartime in an economy and the leftover social and economic infrastructures; second, the nature of the data and variables augmented in the analytical models. Empirical studies found that slow economic growth and low per capita incomes are robustly connected to civil war in an economy (see, for instance, [1]). Again, there is no general consensus on the most effective policies to avoid conflicts or endorse postwar repossession. The present study, which is based on country-specific data—Afghanistan, assesses both the concurrent and enduring effects of war on the economy, following the fundamental concept of war effects on economic performance. Therefore, this triggers the formulation of three key questions, among all others. First, do economic performance and the war effects—both concurrent and residual effects—move together? Second, what would be the scale and magnitude of the war effects on the economy, i.e., monotonic or non-monotonic? Third, is there any causality relationship between war and the economy? Providing consistent and evidence-based answers to these questions will not only help policymakers to consider the asymmetries of the war's effects on the economy when developing and implementing relevant policies; they will also assist future studies to build upon the literature.

Although there are numerous studies concerning the effects of war on various socioeconomic indicators in other countries, such as Libya [10–12]; Syria [13, 14]; Croatia [15, 16]; Sri Lanka [17–19]; Rwanda [2]; and Ukraine [20, 21]; yet studies are rare or even non-existent to articulate the scale and magnitude of the effects of war on the Afghan economy. This significant gap in the literature has caused two key shortcomings. First, the analytical reports and

studies (see, for instance, [22–24]) have been qualitative, producing descriptive information on the status of war and its presumable impacts on the economy, leading to perplexing results and confounded conclusions. Second, the non-existence of comprehensive and complexed modeling empirical works, resulted in knowing little about the scale and magnitude of the effects of long-run war on the economic performance in Afghanistan. Thus, the existing gap—linked to the key questions—directs the study to develop and test three key hypotheses. $H_1$: There is a long-run link between war and the economy; $H_2$: Long-run war has a non-monotonic and asymmetric impact on the economy in the short and long run; and $H_3$: In Afghanistan, there is an asymmetric causal link between long-run war and the economy.

The present study is unique in context, method, and generalizability. It contributes to the existing literature on the war-growth nexus in four ways. First, to the best of the authors' knowledge, this is the first study in the context of a long-running war in Afghanistan, allowing quantitative analysis to determine the scale and magnitude of the effects of war on the Afghan economy. Second, it employs a set of well-known predictors of war and control variables selected by the PCA (principal component analysis) method to provide consistent and accurate estimation. Third, the asymmetric ARDL model is used in this article to test the effects of the predictors on the economy and compares their asymmetric significance with respect to the short- and long-run responses of growth to war. Fourth, it uses asymmetric causality techniques to delve into the causality nexus between predictors to find out any multidimensionality and interdependence between war and the economy over the period of this study. The critical findings of the study are also fourfold. First, it notes that the effects of war are asymmetrically bound with the economy in the long-run. Second, the findings confirm the asymmetric effects of war on the economy both in the short and long runs, while criticality suggests that effects are non-monotonic by scale in the runs. Third, corresponding with the theoretical concept, the non-monotonic impact of war describes higher intensity in the long-run than short-run effects. Fourth, during the period under study, the findings confirm bidirectional causality among the predictors.

The remaining parts of the study are structured as follows. Section two presents a brief literature review both on the theoretical background and the empirical impact of war on the economy. Section three presents the data and describes the variables used in the study. Section four explains the methodology used to analyze the data. Section five presents the results and discusses the findings. Section six concludes the study. Finally, the concluding paragraphs will discuss the policy implications of the findings on the impact of the long-run war on the Afghan economy and provide some relevant policy recommendations.

## 2 Literature review

The existing literature vastly defines war as an intense armed conflict between governments, states, nations, and groups of people who try to achieve certain economic, military, and political benefits that are not available through peaceful discourse [25]. The successful conclusion of an armed conflict may satisfy the desires of a party or a group of insurgents, but it leaves the war-effected nation with severe negative consequences, either concurrently or as residuals after the war ends [26]. According to Serneels and Verpoorten [2], critical gaps will remain in the academic literature to understand the clear and well-supported evidence on the economic consequences of war; what is clear is the substantial economic cost of rehabilitation and reconstruction in a post-conflict environment, along with other psychologically negative consequences that take a long time to recover. It is well-evident that war, terrorism, and organized crimes are interconnected with each other, implying the impact of war on enabling illicit enterprises. For instance, the long-run war in Afghanistan created an open war economy,

affecting both the economic and political stability of Afghanistan and the surrounding regions. As a result, it has become not only the world's largest opium producer and a center for arms dealing but also supports a multibillion-dollar trade in goods smuggled from Dubai to Pakistan [27]. It further facilitated a criminalized economy that financially supports both anti-government insurgents and their adversaries, posing a significant threat to the economic and political sustainability of a country that is already unstable [28]. An extensive empirical literature indicates that the effects of war and armed conflicts on socioeconomic indicators have been critically analyzed in other economies. For instance, in the context of Sri Lanka, Arunatilake et al. [29] examined the effects of war on the economy and found that since 1983, the cost of war was approximately equivalent to Sri Lanka's GDP in 1996.

In a study of the economic repercussions of armed conflicts in the Basque country, Abadie and Gardeazabal [30] discovered that with the outbreak of terrorism in the 1960s, per capita GDP declined by 10% compared to synthetic control regions without violence. Furthermore, the authors discovered that stocks of companies doing business in the Basque region had a positive relative performance when the cease-fire became credible and a negative relative performance after the cease-fire ended. Kang and Meernik [31] examined the effects of war on economic growth using the two-stage ordinary least square method and a set of panel data from 1960–2002. They found that civil war has a significantly negative impact on economic growth and that it reduces the power of special interests, disrupts technological innovation, and human capital accumulation (see also, [32]).

Gates et al. [33] assessed the effects of war on economic growth and discovered that conflict has clear damaging effects on poverty and hunger, education, child mortality, and access to drinkable water. They further found that a medium-sized armed conflict of 2500 battle deaths would increase malnutrition by 3.3%, decrease life expectancy by almost 1 year, increase child mortality by 10%, and divest 1.8% of the population from access to clean water. Moreover, Camacho and Rodriguez [34] employed two panel data sets to test the causal effects of the armed conflict on the firms' exit in Colombia, using the fixed effect model for estimation. The authors found that one standard deviation shock would increase the number of insurgent and guerrilla attacks in a municipality and the probability of plant exit by 5.5 percentage points. The effect is robust for new manufacturing plants, with a smaller number of workers and low levels of capital. Similarly, Ali [35] examined the economic cost of the Darfur conflict and discovered that the cost of war has been equivalent to 171% of Sudan's 2003 GDP in Darfur, implying a clear cut of spending from other macroeconomic sectors. In addition to a substantial number of deaths and injuries, the war had a significantly negative impact on the country's economy during wartime.

Ganegodage and Rambaldi [36] employed the neoclassical endogenous theoretical framework to test the impact of civil war on Sri Lanka's economic growth and found that war has significantly adverse effects on the economy, proxied by GDP both in the short and long runs. They further show that high returns from investment in technological and physical capital do not explain sizable positive outwardness. Only short-term significant effects of economic openness on growth are found to be significant. Furthermore, Serneels and Verpoorten [2] examined the impact of civil war in Rwanda using a set of micro-data for the early 1960s and considered the endogeneity effects of the violence. The authors found that the households and localities that experienced the most intense armed conflict are lagging behind in terms of consumption six years after the armed conflict. The authors also discovered that returns from land and labor were significantly different between localities experiencing low and high intensities of armed conflict.

Kutan and Yaya [37] analyzed the effects of war on financial and economic risk in Colombia using a set of data spanning from 2002–2012. In the presence of a large-scale terrorist

attack, the authors examined the abnormal returns in the stock market and found that abnormal returns in the stock market significantly responded to the terrorist attacks in Colombia. Their findings also show that such attacks increase the volatility of the abnormal returns while the overall economic performance suffers statistically.

Aziz and Asadullah [38] investigated the causality relationships and the impact of military expenditure on economic growth for the period 1990–2013 using a large panel dataset for 70 developing economies. The authors employed the generalized method of moments, fixed effects, and random effects models by regression. Considering the non-linearity assumptions, their findings indicate a significant effect of military expenditure on economic growth. While controlling for conflict, the effect of military expenditure conditional upon conflict exposure was also positive and statistically significant. Importantly, the authors found a joint effect of armed conflict and military expenditure on economic growth. Moreover, Bove et al. [39] explored the heterogenous effects of a civil war on the economy, proxied by GDP, using a case study of large panel data and a synthetic control method. The authors provide statistical evidence that civil war, on average, has a significantly negative effect on the economy, while little evidence was found to show its enduring effects. The authors also provide a counter-example of some economies that highlight positive reactions towards armed conflicts.

Torres-Garcia et al. [40] analyzed a set of panel data from 1980 to 2015 comprising 55 countries to test the nexus between aggregate savings rates and war intensity during the armed conflict period. Their results reveal that economies that have suffered some type of conflict show a 2.7% decline in the saving rate on average compared to those that have not suffered such conflicts. The authors show that if there is a higher intensity of conflict, the saving rates would decrease by 2.5% in war-torn economies compared with those having a lower intensity of armed conflict. They also highlight that a nonlinear nexus exists between saving rates and conflict duration, signifying that the effects of armed conflict on savings decrease with the laps of time. Similarly, Hodler [41] examined the effects of war and genocides on the economic development in Rwanda using the synthetic control method. Controlling for other predictors, the author found that the GDP in Rwanda declined by 58% in 1994, corresponding to a sharp decline in the per capita GDP by 31% considering the death of approximately eight hundred thousand people. The counterfactual results of the study show that Rwanda had a quick recovery, experiencing only short-run war effects, with evidence indicating that the agriculture sector of the country was less severely hit by the genocide than those of industry and service sectors.

Frolov and Bosenko [42] analyzed the financial and legal resistance of enterprises during wartime, providing contextual examples from the Gaza Strip and Syria. The authors discovered that, while war has a significant impact on economic performance, which declines during the war and thereafter until the economy recovers, enterprises have learned to exist in unstable conditions, forming new strategies and reactions to events. Moreover, in a comparative study by Bataeva et al. [43], who assessed the effects of war on investment attractiveness, the authors highlight that the effects of civil wars in Latin America, African countries, Iraq, and Syria significantly differ at macro and microeconomic levels. Throughout, the authors argue, conflict-zone economies substantially relinquish their investment attractiveness, resulting in the economy showing a sharp decline in capital investments.

Heger and Neumayer [44] evaluated the legacy effects of violence on the gross domestic product of the province of Aceh in Indonesia using subnational district-level data. They found that armed conflict had a residual negative impact on the economy, though a general peace dividend existed. They also provide evidence that districts with lower conflict intensity had swifter growth during 2005 than those with higher intensity. The authors argue that the effects of conflict have been short-term, with no statistical evidence of their long-term impact. Finally, Bluszcz and Valente [45] examined the causal impact of the Donbass war on the economy of

Ukraine using cross-country data spanning from 1995–2017. The authors employed a synthetic control method and found that the per capita GDP of Ukraine that has foregone due to the war rounded to 15.1% during the period from 2013 to 2017, while during the period from 2013 to 2016, the war-effected regions, such as Donetsk and Luhansk, show a causal effect of 47%.

The existing literature on the effects of war on economic growth is rarely straightforward, and anyone is likely to be overwhelmed by the number of pertinent studies that have been completed to date. This limitation is more highlighted by the fact that the analysis of the effects of war has only undergone qualitative and descriptive analysis by the publication of various reports referenced to international debates, while little attention is paid to this phenomenon as scholarly published documents in Afghanistan. A review of the existing literature offers some sense of what is known as war effects, but one of the consequences of armed conflict that is equally revealing is how little is known about the scale and magnitude of its effects on the socioeconomic predictors with any degree of certainty. Therefore, the scarcity of scholarly studies concerning the scale and magnitude of the war effects on the economy in Afghanistan and the war-torn countries sharing the same nature makes the present study important and relevant to fill the existing gaps in the literature.

## 3 Data and variables

As per the availability of data, this study uses a set of quarterly time-series data spanning from 2002Q3–2020Q4 relevant to Afghanistan's war and economic predictors. Based on two key motives, Afghanistan is selected as the context of this study. First, it is a true representation of history's longest war and an example of multiple state failures, much of which is the outcome of ideological battles. Second, there is a scarcity of comprehensive and quantitative studies that provide empirical insights into the scope and magnitude of the effects of war on the Afghan economy. The datasets have been collected from WDI (World Development Indicators), sources relevant to the World Bank and the US Department of Defense Budget FY2002–FY2020. The variables used in the study are consistent with the theoretical foundation and recent empirical studies and include (i) per capita GDP, (ii) per capita energy consumption, (iii) per capita final household consumption expenditure, (iv) per capita remittance, (v) per capita cost of war, (vi) per capita capital investment proxied by gross fixed capital formation, (vii) per capita foreign direct investments, (viii) population growth rate, and (ix) inflation rate. Variable (i) is expressed in hundreds of US dollars, variables (ii–iv) are expressed in thousands of 2015 constant US dollars, variables (v–vii) are expressed in millions of 2015 constant US dollars, and variables (viii and ix) are expressed as annual percentages. Table 1 provides complete information on the variables' descriptions, symbols, measurement, and sources from which they are collected.

Per capita GDP is used as the dependent variable to measure the variability of economic performance and growth, as well as inflation, to assess the extent of macroeconomic instability in the country due to the impact of war components [46]. For instance, theory predicts that war clutches the production factors and thus the aggregate output of an economy declines [47], so the economy may be squeezed during wartime. The cost of war is used to measure the amount of money spent by the United States in Afghanistan from 2002 to 2020. Due to lack of availability of the data, the cost of war presents aggregate data for Afghanistan and is not disaggregated by province and sector, although the war intensity has been higher in some provinces than in others during the period of this study. Furthermore, this proxy has two more features than other proxies used by recent studies. First, it allows a more accurate estimation than the number of people killed and injured. Second, it provides actual data on the amount of money spent on operating pure military operations.

**Table 1. Description of variables.**

| Full name | Symbols | Measurement | Sources |
|---|---|---|---|
| **Dependent variable** | | | |
| Per capita GDP | PGDP | Thousands of 2015 constant US dollars | WDI, the World Bank Group |
| **War variable** | | | |
| Per capita cost of war | PCOW | Millions of US dollar | The US Department of Defense Budget FY02–FY18 |
| **Control variables** | | | |
| Per capita final household consumption expenditure | PFCE | Millions of US dollars | WDI, the World Bank Group |
| Per capital energy consumption | PCEC | Thousands of US dollars | World Data (Energy consumption) |
| Per capita remittance | PCR | Thousands of US dollars | WDI, the World Bank Group |
| Per capita capital investment | PCCI | Millions of US dollars | WDI, the World Bank Group |
| Per capita foreign direct investment | PFDI | Millions of US dollars | WDI, the World Bank Group |
| Population growth rate | PGR | Annual (%) | WDI, the World Bank Group |
| Inflation rate | INR | Annual (%) | WDI, the World Bank Group |

Notes: WDI: World Development Indicators, FY: Financial Year, US: United States. Sample size adjusted from 2002Q3–2020Q4.

For the control variables, capital investment proxied by gross fixed capital formation measures the overall investment of an economy in the acquisition of assets and production infrastructure [48]. Since armed conflicts force governments to reallocate budgets, capital investment is assumed to swiftly react to the reallocation during wartime and thus affect the overall economic performance in a conflict environment. Both economic theories and empirical studies show that there exists a direct link between household consumption expenditures and economic growth [49] and it is used to control its effects on economic growth. Moreover, studies by Dogan and Turkekul [50], Magazzino [51], and Ma and Fu [52] suggest employing the energy consumption when testing the effects of other predictors on economic growth to avoid any potential specification bias. Based on the inflation-targeting regime of the central bank of Afghanistan, it seriously attempted to control the inflationary impact on economic growth during wartime. Therefore, along with the war predictors, it is important to capture the inflationary effects on economic growth (see, for instance, [53]. Finally, population growth rate is used to measure the annual change in the population and is employed to capture its effect on growth. Literature shows that population growth has significant effects on economic growth either directly or indirectly [54]. In sum, the variables have been thoroughly selected to provide consistent results and avoid omitted variable bias, though some recent studies employed different control variables, such as climate changes, $CO_2$ emissions, and financial crises. To validate the predictors, the present study computes the PCA (principal component analysis) to test the explanatory power of the variables and reports the results in Table 2.

Table 2 reports the specific computation by PCA, demonstrating that the first to nine components have explanatory powers ranging from 66 percent to 0.01 percent. This means that the first component accounts for 66 percent of the variation in the predictors, while the last component accounts for less than 1 percent. It concludes that the first component has the greatest explanatory power to translate the variations when compared to the other variables. It is important to note that the estimates in the fifth column are used to weight the war and control indicator fed into the subsequent regressions.

## 4 Methods

Based on the key objectives of this study, this section explains the methods used to assess the asymmetric effects of the long-run war on economic growth in Afghanistan, using the

**Table 2. War-growth aggregation index.**

| Components | Eigen-values | Variance (%) | Cumulative Proportion (%) | First principal component |
|---|---|---|---|---|
| 1 | 4.855 | 0.662 | 0.662 | -0.175 [INR] |
| 2 | 1.642 | 0.105 | 0.767 | 0.400 [PCCI] |
| 3 | 1.055 | 0.093 | 0.860 | -0.393 [PCEC] |
| 4 | 0.741 | 0.074 | 0.934 | 0.306 [PCOW] |
| 5 | 0.527 | 0.033 | 0.967 | 0.324 [PCR] |
| 6 | 0.479 | 0.021 | 0.988 | -0.018 [PFDI] |
| 7 | 0.421 | 0.007 | 0.995 | 0.438 [PGDP] |
| 8 | 0.408 | 0.004 | 0.999 | -0.269 [PGR] |
| 9 | 0.334 | 0.001 | 1.000 | 0.438 [PHFC] |

Notes: Eigenvalues: (sum = 9, average = 1). Sample size adjusted from 2002Q3–2020Q4. [] presents the predictors' weigh. INR = Inflation rate, PCCI = Per capita capital investment, PCEC = Per capita energy consumption, PCOW = Per capita cost of war, PCR = Per capita remittance, PFDI = Per capita foreign direct investment, PGDP = Per capita GDP, PGR = population growth rate, PHFC = Per capita household final consumption expenditure.

proposed model by Shin et al. [55], that is—the non-linear autoregressive distributed lags method. However, following certain econometric literature, the specification begins with pre-requisite tests of stationarity, cointegration, and test of symmetries to facilitate appropriate model selection. Therefore, the following sub-sections explain the series of sequential tests:

## 4.1 Unit root tests

In time-series analysis, it is important to begin the analysis with determining the integrating order of the predictors to avoid any potential misspecification and fabricated results. In this faith, the Augmented Dickey and Fuller [56], Phillips and Perron [57], and Kwiatkowski et al. [58] methods are used. Augmented Dickey and Fuller (ADF) and Phillips and Perron (PP) methods are employed to test the null of $H_O$: $\delta = 0$ (non-stationarity) vs. $H_A$: $\delta \neq 0$ (stationarity), where the KPSS tests the null of $H_O : \sigma_\varepsilon^2 = 0$ vs. $H_A : \sigma_\varepsilon^2 > 0$. The ADF, PP, and the KPSS equations used in the study take the following forms, respectively:

$$\Delta x_t = \varphi + \eta T + \lambda x_{t-1} + \sum_{i=1}^{q-1} \delta_1 \Delta x_{t-1} + \varepsilon_t \tag{1}$$

$$\Delta x_t = \varphi + \eta t + \delta x_{t-1} + \varepsilon_t \tag{2}$$

$$x_t = \varphi + \delta(r_t) + \eta t + \varepsilon_t \sim WN(0, \sigma_\varepsilon^2) \tag{3}$$

where the change sign $\Delta$ is the first difference operator, $\varphi$ presents the intercept, $\mu$ is the time trend coefficient, $\delta$ is the coefficient of the variable being tested for unit root, and $\varepsilon$ presents the error term of the models 1–3. In Eq (3), $(r_t)$ is the random walk process. To compute Eqs (1)–(3), the optimal lag length is automatically selected using the AIC (Akaike information criterion), SIC (Schwarz information criterion), and HQIC (Hannan-Quinn information criterion) frameworks.

## 4.2 Cointegration test

Considering the formulated hypotheses and to establish the long-run nexus amid war and growth predictors, this study employs the ARDL bound test of Pesaran et al. [59]. For brevity, let $y_t$, $x_t$, and $z_t$ be a set of variables presenting economic growth, war predictor, and the control

variables, respectively, the ARDL bound test to cointegration can be expressed as:

$$\begin{aligned}
\Delta y_t &= \varphi + \xi_1 y_{t-1} + \xi_2 x_{t-1} + \xi_3 z_{t-1} + \vartheta k_t \\
&+ \sum_{i=1}^{p} \lambda_{1i} \Delta y_{t-1} + \sum_{i=0}^{q} \lambda_{2i} \Delta x_{t-1} + \sum_{i=0}^{q} \lambda_{3i} \Delta z_{t-1} + \varepsilon_t
\end{aligned} \tag{4}$$

where all the variables are described before, $\varphi$ is the intercept, $\lambda(\xi)$ presents the short-run (long-run) coefficients, $\vartheta$ is the trend regressor, and $\varepsilon$ is the stochastic error term of the model. Eq (4) is cointegrated if it rejects the null hypothesis of $\xi_1 = \xi_2 = \xi_3 = 0$ in the favor of $\xi_1 \neq \xi_2 \neq \xi_3 \neq 0$ jointly or separately as $\xi_1 = 0$, $\xi_2 = 0$, and $\xi_3 = 0$, using F-statistics [60]. If the F-statistics are greater than the upper bound I(1) critical value of a desired level, the null is rejected, while it fails to reject the null if the F-statistics are less than the lower bound I(0) critical value. In case the F-statistics fall between the lower bound and the upper bound critical values, the test is inconclusive about the null hypothesis [61]. The use of ARDL bound test has various advantages over other common cointegration methods. First, it allows the predictors to follow mixed integrating orders. Second, it provides consistent and accurate results even in small samples [62]. Third, it allows the dependent and independent variables to augment different lags in computation.

## 4.3 Non-linear ARDL model

Assuming that all the indicators follow mixed integrating orders of I(0) and I(1) without any I(2) series, the study proceeds to estimate the asymmetric ARDL model to test the short and long-run effects of the war and other control variables on economic growth, using the non-linear ARDL model of Shin et al. [63]. The asymmetric ARDL model allows the decomposition of the partial sums of both the positive and negative squares to explore the non-linearities in the short and long-run effects. To initiate the asymmetric ARDL modeling, the present study modifies Eq (4) and represents it as:

$$y_t = \lambda_{1t}^+ x_t^+ + \lambda_{2t}^- x_t^- + \lambda_{3t}^+ z_t^+ + \lambda_{4t}^- z_t^- + \mu_t \tag{5}$$

where the variables hold the same meaning as defined in Eq (4), $\lambda_t^+$ and $\lambda_t^-$ present the positive and negative changes in $x_t$ and $z_t$, respectively (see, for instance, [63]). The positive and negative changes are incorporated into the model by the function $x_t = x_t + x_t^+ + x_t^-$ and $z_t = z_t + z_t^+ + z_t^-$ utilizing the following process as:

$$\begin{aligned}
x_t^+ &= \sum_{j=1}^{t} \Delta x_j^+ = \sum_{j=1}^{t} \max(\Delta x_t, 0), \quad x_t^- = \sum_{j=1}^{t} \Delta x_j^- = \sum_{j=1}^{t} \min(\Delta x_t, 0) \\
z_t^+ &= \sum_{j=1}^{t} \Delta z_j^+ = \sum_{j=1}^{t} \max(\Delta z_t, 0), \quad z_t^- = \sum_{j=1}^{t} \Delta z_j^- = \sum_{j=1}^{t} \min(\Delta z_t, 0)
\end{aligned} \tag{6}$$

The linear I(0) combination in Eq (5) and non-linear partial sum of squares are as:

$$d_t = k + \vartheta_{1t}^+ y_t^+ + \vartheta_{2t}^- y_t^- + \omega_{2t}^+ x_t^+ + \omega_{2t}^- x_t^- + \omega_{3t}^+ z_t^+ + \omega_{4t}^- z_t^- + e_t \tag{7}$$

Thus, Eq (7) would be stationary if $d_t = I(0)$ and with asymmetric long-run cointegration rejecting the null of $\vartheta_t^+ = \vartheta_t^- = \lambda_t^+ = \lambda_t^- = 0$ in favor of its alternative $\vartheta_t^+ \neq \vartheta_t^- \neq \lambda_t^+ \neq \lambda_t^- \neq 0$. Eqs (5) and (7) may assume endogeneity and multicollinearity problems and that can be fixed by integrating the dynamic forms of the equations as:

$$y_t = \sum_{i=1}^{p} \eta u n_{t-1} + \sum_{i=0}^{q} \left( \gamma_1^+ x_{t-i}^+ + \gamma_2^- x_{t-i}^- + \gamma_3^+ z_{t-i}^+ + \gamma_4^- z_{t-i}^- \right) + u_t \tag{8}$$

where $\eta$ presents the autoregressive parameter, $\gamma$ is the dynamic parameter of the model adjusting the dynamic format of cointegration. Thus, considering this, the asymmetric ARDL

model used in the present study takes the following form:

$$\Delta y_t = \rho y_{t-i} + \xi_1^+ x_{t-i}^+ + \xi_2^- x_{t-i}^- + \xi_3^+ z_{t-i}^+ + \xi_4^- z_{t-i}^- + \sum_{i=1}^{p} \phi_i \Delta y_{t-i} + \sum_{i=0}^{q} \lambda_{1i}^+ \Delta x_{t-i}^+$$
$$+ \sum_{i=0}^{q} \lambda_{2i}^- \Delta x_{t-i}^- + \sum_{i=0}^{q} \lambda_{3i}^+ \Delta z_{t-i}^+ + \sum_{i=0}^{q} \lambda_{4i}^- \Delta z_{t-i}^- + e_t \tag{9}$$

where all the variables are defined before, $\xi_t^+ (\xi_t^-)$ are the long-run non-linear, say, positive (negative) coefficients, $\lambda_t^+ (\lambda_t^-)$ short-run positive (negative) coefficients. Moreover, the present study investigates that how in the long-run, the per capita GDP, therefore, economic growth responds to a dynamic asymmetric shock by the war and other control variables, using the dynamic multipliers approach. The dynamic multiplier is used to expedite the sequential growth element as it changes from milieus of the previous short-run dynamism and the early non-stabilities into a new equilibrium after a standard shock. The equation used is expressed as:

$$mh^+ = \sum_{i=0}^{h} \frac{\partial(y_t)}{\partial(x_t^+)} = \sum_{i=0}^{h} \varphi_i^+, \quad mh^- = \sum_{i=0}^{h} \frac{\partial(y_t)}{\partial(x_t^-)} = \sum_{i=0}^{h} \varphi_i^-,$$
$$mh^+ = \sum_{i=0}^{h} \frac{\partial(y_t)}{\partial(z_t^+)} = \sum_{i=0}^{h} \varphi_i^+, \quad mh^- = \sum_{i=0}^{h} \frac{\partial(y_t)}{\partial(z_t^-)} = \sum_{i=0}^{h} \varphi_i^-, \tag{10}$$

where $mh^+(mh^-)$ are the asymmetric long-run coefficients and are empirically consistent when $m$ tends to infinity and they are important to preserve the vital evidence responsible for the macroeconomic volatilities.

## 4.4 Asymmetric causality test

Finally, this study investigates the causal relationships among indicators. In this faith, it applies the asymmetric causality tests of Hatemi-J [64], which is based on Toda-Yamamoto's [65] approach. The study initiates building the modified vector autoregressive $(K + d_{max})$ approach based on the optimal lag selected using the AIC, SIC, and HQIC frameworks and specifies the Toda-Yamamoto pairwise equations that can take the following forms:

$$y_t = \varphi + \left( \sum_{i=1}^{k} \vartheta_{1t} x_{t-1} + \sum_{i=k+1}^{d_{max}} \vartheta_{2t} x_{t-2} \right) + \left( \sum_{i=1}^{k} \phi_{1t} x_{t-1} + \sum_{i=k+1}^{d_{max}} \phi_{2t} x_{t-2} \right) + e_t$$
$$x_t = \delta + \left( \sum_{i=1}^{k} \omega_{1t} x_{t-1} + \sum_{i=k+1}^{d_{max}} \omega_{2t} x_{t-2} \right) + \left( \sum_{i=1}^{k} \eta_{1t} x_{t-1} + \sum_{i=k+1}^{d_{max}} \eta_{2t} x_{t-2} \right) + e_t \tag{11}$$

where $\varphi$ and $\delta$ are the intercepts, $k$ is the lag length automatically selected via AIC, SIC, and HQIC methods, and $(K + d_{max})$ is the number of cointegrating orders of the predictors. Thus, let the positive and negative shocks of each predictor in a cumulative form be $y_{1t}^+ = \sum_{i=1}^{t} \varepsilon_{1t}^+$ and $y_{1t}^- = \sum_{i=1}^{t} \varepsilon_{1t}^-, y_{2t}^+ = \sum_{i=1}^{t} \varepsilon_{2t}^+$ and $y_{2t}^- = \sum_{i=1}^{t} \varepsilon_{2t}^-$ with a permanent effect on the underlying indicator. Thus, $y_t^+ = (y_1^+, y_2^+)$ and $y_t^- = (y_1^-, y_2^-)$ vectors are employed to test for the asymmetric causality nexus in the following vector autoregressive model:

$$y_t^+ = \nu + A_1 y_{t-1}^+ + \ldots + A_p y_{t-p}^+ + u_t^+ \tag{12}$$

where $\nu, y_t^+, u_t^+$, and $A$ present the 2×1 vector of the intercept, 2×1 vector of the predictors, 2×1 vector of the error term, and the 2×2 matrix of parameters for $k$ ($k = 1, 2, 3, \ldots, p$) lag orders, respectively. Based on the Toda and Yamamoto [65] approach, the Wald test following asymptotic chi-squared distribution is applied to test the null of no asymmetric causal relationship between the variables. Moreover, to control for the abnormal and ARCH (autoregressive conditional heteroskedasticity) effects in the residual, the present study employs bootstrap

simulation with 1,000 replications to find the critical values. Eq (12) is capable to capture the upside and downside causality nexus between economic growth and the war predictors, assuming that the economic growth is asymmetrically affected by the predictors and that it reacts more to both positive and negative shocks from the war. In such a context, common methods fail to capture the in-depth causal nexus between the indicators, while an asymmetric causality test accounts for different positive and negative asymmetrical causal effects of the predictors on the outcome variable—that is, economic growth [65]. Finally, the study computes some important post-estimation diagnostic tests to ensure the results are robust and accurate.

## 5 Results and discussions

### 5.1 Unit root analysis

The analysis begins with the unit root tests of the ADF, PP, and KPSS methods, using both constant and trend regressors. The estimation is based on the optimal lag length selection via AIC and SIC approaches. The results are reported in Table 3 and indicate that PCEC (per capita energy consumption) and PCCI (per capita capital investment) are level-stationary variables, implying that their *p-values* are significant for the rejected null of non-stationarity at 1% level by all three methods. The test statistics for the other variables, such as PGDP, PCOW, PFCE, PCR, PFDI, PGR, and INR, are insignificant to reject the null at the level; instead, all three methods reject it at the first difference. Therefore, the results reveal that PCEC and PCCI follow the I(0) series while the remaining predictors follow the I(1) series, leading the study to proceed with the estimation of the bound test to examine any long-run nexus among them. Moreover, the unit root results support the model specification discussed earlier, and thus, it computes both symmetric and asymmetric ARDL models, that are appropriate for mixed integrating orders of I(0) and I(1) without any I(2) series [66] and [67].

### 5.2 Cointegration analysis results

Next, to examine the long-run nexus amid growth, war, and control predictors, both symmetric and asymmetric ARDL bound tests are applied. The bound test estimation includes

**Table 3. Unit root test results.**

| Variables | Augmented Dickey-Fuller | | Phillips-Perron | | KPSS | |
|---|---|---|---|---|---|---|
| | I(0) | I(1) | I(0) | I(1) | I(0) | I(1) |
| PGDP | -2.152 | -4.486*** | -2.018 | -4.247*** | 0.969*** | 0.327 |
| PCOW | -2.138 | -5.852*** | -1.998 | -4.854*** | 0.931*** | 0.118 |
| PFCE | -2.095 | -3.634*** | -1.871 | -4.021*** | 0.874*** | 0.289 |
| PCEC | -4.062*** | -4.997*** | -3.974*** | -4.282*** | 0.344 | 0.801*** |
| PCR | -0.266 | -4.313*** | -1.122 | -3.898*** | 1.011*** | 0.302 |
| PCCI | -3.840*** | -4.991*** | -4.011*** | -4.447*** | 0.289 | 0.899*** |
| PFDI | -1.465 | -3.982*** | -1.377 | -4.099*** | 0.884*** | 0.149 |
| PGR | -0.993 | -3.849*** | -0.826 | -3.689*** | 1.128*** | 0.129 |
| INR | -1.044 | -4.038*** | -1.031 | -4.011*** | 0.861*** | 0.133 |

Notes:

***, **, and * present significance at 1%, 5%, and 10%, respectively.

PGDP = Per capita GDP, PCOW = Per capita cost of war, PFCE = Per capita final household consumption expenditures, PCEC = Per capita energy consumption, PCR = Per capita remittance, PCCI = Per capita capital investment, PFDI = Per capita foreign direct investment, PGR = Population growth rate, INR = Inflation rate, KPSS = Kwiatkowski, Phillips, Schmidt, and Shin. Critical values for KPSS at 1% and 5% are 0.739 and 0.463, respectively.

**Table 4. Bound test results.**

| Statistics | Symmetric ARDL bound test | | | Asymmetric ARDL bound test | | |
|---|---|---|---|---|---|---|
| | Values | Critical values (1%) | | Values | Critical values (1%) | |
| | | I(0) | I(1) | | (0) | I(1) |
| F-statistics | 1.544 | 2.62 | 3.77 | 18.339*** | 2.73 | 3.90 |
| t-statistics | -1.015 | -3.39 | -3.97 | -21.044*** | -3.31 | -3.97 |

Notes:

***, **, and * present significance at 1%, 5%, and 10%, respectively.

K = 8 and implies the number of regressors augmented in the models.

automatic lag selection of ($p = 2$, $q = 2$) for dependent and independent variables via AIC, SIC, and HQIC methods. The results are reported in Table 4 demonstrate that economic growth is only asymmetrically bounded with the war and control variables in the long-run (F = 18.339, t = -21.044) being greater than the critical values rejecting the null of $\rho = \xi_1^+ = \xi_2^- = \xi_3^+ = \xi_4^- = 0$ at the 1% significant level, while the results of the symmetric ARDL bound (F = 1.544, t = -1.055) are insignificant to reject the null of no cointegration, say, $H_O$: $\xi_1 = \xi_2 = \xi_3 = 0$ at either of the significant levels. This implies that the war-growth nexus is only asymmetrically bound in the long-run, which triggers further insights into their short and long run asymmetries (see, Table 5). The findings are related to the fact that it is reasonable to assume that long-run war and the relevant predictors are linked to economic growth and the tendency to move together in the long run. According to the theoretical foundation, long-term wars, such as those in Afghanistan that lasted more than four decades, imply two types of tie-ups, such as concurrent impact and residual effects, which move in tandem with socioeconomic indicators.

Based on the critical result of the asymmetric ARDL bound test reported in Table 4, this study computed the Wald statistics to test the null hypothesis of short-run and long-run symmetries. The results reported in Table 5 confirm that the non-linear relationships between economic growth, war, and control predictors by rejecting the null of short and long run symmetries, say, $H_O : \xi_t^+ = \xi_t^- \sim \chi^2$ and $H_O : \lambda_t^+ = \lambda_t^- \sim \chi^2$ at the 1% significant levels. The finding reveals that the war and other control variables' positive and negative partial sums of squares differently impact the per capita GDP—the economic growth in the short and long runs, while it also suggests exploring the scale and sign of the effects.

### 5.3 Asymmetric ARDL estimates

Considering the specified methodology and based on the outcome of the asymmetric long-run nexus and the rejected null of both short and long-run symmetries (see Tables 4 and 5), the study proceeds to estimate the asymmetric ARDL model using Eq (9). It further computes and presents the results of the dynamic multipliers using Eq (10) to delve into the asymmetric

**Table 5. Wald test for symmetries.**

| Wald test statistics | Chi-squared statistics | p-values |
|---|---|---|
| Short-run symmetries, $\sum_{j=0}^{p-1} n_j^+ = \sum_{j=0}^{p-1} n_j^-$ test statistics | 1309.478*** | 0.000 |
| Long-run symmetries, $-\xi^+/\varphi = -\xi^-/\theta$ test statistics | 18294.005*** | 0.000 |

Notes:

***, **, and * present significance at 1%, 5%, and 10%, respectively.

**Table 6. Asymmetric ARDL estimates.**

| Statistics | Model estimated: NARDL | | | | | | | |
|---|---|---|---|---|---|---|---|---|
| | $PCOW_t^+$ | $PCOW_t^-$ | $PFCE_t^+$ | $PFCE_t^-$ | $PCEC_t^+$ | $PCEC_t^-$ | $PCR_t^+$ | $PCR_t^-$ |
| Coefficients | -30.65*** | 10.37*** | 56.75*** | -60.98*** | -8.11*** | 2.66*** | -0.72*** | 6.67*** |
| t-statistics | -8.254 | 3.719 | 6.339 | -4.841 | -5.011 | 9.141 | -4.155 | 4.833 |
| p-values | 0.000 | 0.009 | 0.000 | 0.000 | 0.000 | 0.000 | 0.000 | 0.004 |
| | $PCCI_t^+$ | $PCCI_t^-$ | $PFDI_t^+$ | $PFDI_t^-$ | $PGR_t^+$ | $PGR_t^-$ | $INR_t^+$ | $INR_t^-$ |
| Coefficients | 19.88*** | -11.95*** | 0.88*** | -2.87*** | -21.42* | 7.04*** | -14.25*** | 4.07*** |
| t-statistics | 5.312 | -6.011 | 4.901 | -3.991 | -3.452 | 8.133 | -5.862 | 5.097 |
| p-values | 0.000 | 0.000 | 0.001 | 0.009 | 0.051 | 0.000 | 0.000 | 0.000 |

Notes:

***, **, and * present significance at 1%, 5%, and 10%, respectively.

PGDP = Per capita GDP, PCOW = Per capita cost of war, PFCE = Per capita final household consumption expenditures, PCEC = Per capita energy consumption, PCR = Per capita remittance, PCCI = Per capita capital investment, PFDI = Per capita foreign direct investment, PGR = Population growth rate, INR = Inflation rate, ECT = Error-correcting term. [+] and [−] present positive and negative partial sum of squares, respectively.

effects of the per capita cost of war and other explanatory variables on the per capita GDP—that is, the economic growth in Afghanistan. The estimation of Eq (9) is based on the optimal lag length using the AIC, SIC, and HQIC frameworks, using the "varsoc" command. Table 6 reports the standard asymmetric ARDL estimates of both positive and negative partial sum effects, while Table 7 reports the short and long-run estimates of the asymmetric ARDL model. The results of post-estimate diagnostic tests relevant to the computation of Eq 9 are provided in the rare parts of Table 7.

The results reported in Table 6 provide important and interesting highlights. They indicate that a positive change in the per capita cost of war decreases the per capita GDP by $30.65 and a negative partial sum change in the per capita cost of war increases the per capita GDP by $10.37. In an empirical sense, as found by estimations, a positive change (negative impact) of war on the economy is higher than its negative change (positive impact), giving rise to an opportunity cost of $10.37 per capita foregone in the case of continued war in Afghanistan. In a theoretical sense, the result is consistent with the concept of the effects of armed conflict and violence on economic growth in a conflict environment (see, for instance, [68, 69]). In line with the practical survey conducted by Catani et al. [70] on the war in Afghanistan, the results confirm that the intensity of war significantly decreases economic and societal performance, while a decline in the intensity of war creates a temporal relaxation, allowing the nation to produce higher output. With respect to the control variables, the results indicate that all predictors asymmetrically effect economic growth. A positive partial change in per capita household consumption expenditure spurs economic growth by $56.75, while its negative partial change reduces the growth by $8.11. Intuitively, household consumption is widely regarded as one of the ultimate goals of economic activity, and the level of per capita consumption is frequently observed as a key indicator of an economy's productive success [71]. In comparison to theoretical expectations, the results confirm that a positive partial sum change in per capita capital investment and per capita FDI raises per capita GDP, while negative partial sum changes lower economic growth. This is also linked with the fact that economic growth swiftly responds to the advancement in technological, physical, and human capital accumulation. Neoclassical theory considers that economic growth and FDI are correlated, implying that increased FDI inflows inject different technologies, expertise, and foreign investment while

**Table 7. Short-run and long-run asymmetric estimates.**

| Variables | Short-run effects | | | Long-run effects | | |
|---|---|---|---|---|---|---|
| | Coefficients | t-statistics | p-values | Coefficients | t-statistics | p-values |
| $PCOW_t^+$ | -10.099*** | -6.371 | 0.000 | -15.149*** | -4.061 | 0.001 |
| $PCOW_t^-$ | 23.436*** | 5.993 | 0.000 | 56.438*** | 6.229 | 0.000 |
| $PFCE_t^+$ | 16.274*** | 11.450 | 0.000 | 23.275*** | 5.761 | 0.000 |
| $PFCE_t^-$ | -4.475*** | -5.811 | 0.000 | -24.771*** | -8.001 | 0.000 |
| $PCEC_t^+$ | -10.937*** | -4.755 | 0.000 | -3.327*** | -4.027 | 0.009 |
| $PCEC_t^-$ | 2.142*** | 6.449 | 0.000 | 10.924*** | 7.111 | 0.000 |
| $PCR_t^+$ | -1.754*** | -10.338 | 0.000 | -2.971*** | -7.854 | 0.000 |
| $PCR_t^-$ | 4.833*** | 5.999 | 0.000 | 2.972*** | 6.997 | 0.000 |
| $PCCI_t^+$ | 15.333*** | 4.559 | 0.000 | 49.017*** | 7.873 | 0.000 |
| $PCCI_t^-$ | -27.667*** | -6.712 | 0.000 | -81.533** | -12.081 | 0.000 |
| $PFDI_t^+$ | 44.853*** | 6.683 | 0.000 | 35.551*** | 5.381 | 0.000 |
| $PFDI_t^-$ | -9.541*** | -5.367 | 0.000 | -11.891*** | -4.644 | 0.003 |
| $PGR_t^+$ | -23.411 | 0.921 | 0.645 | -20.318*** | -5.881 | 0.000 |
| $PGR_t^-$ | 16.984 | 1.834 | 0.553 | 11.842*** | 4.825 | 0.008 |
| $INR_t^+$ | -10.851*** | -11.951 | 0.000 | -16.023*** | -4.784 | 0.000 |
| $INR_t^-$ | 5.773*** | 6.671 | 0.000 | 10.988*** | 9.556 | 0.000 |
| Diagnostic checks | | | | | | |
| Adjusted r-squared | 0.849 | | | | CUSUM | Stable |
| F-statistics [20, 50] | 23.448*** | | | | CUSUMSQ | Stable |
| Portmanteau [chi$^2$] | 0.861 | | | | | |
| Breusch-Pagan heteroskedasticity test [chi$^2$] | 1.009 | | | | | |
| Ramsey RESET [F] | 2.101 | | | | | |
| Jarque-Bera [chi$^2$] | 1.412 | | | | | |

Notes:

***, **, and * present significance at 1%, 5%, and 10%, respectively.

PGDP = Per capita GDP, PCOW = Per capita cost of war, PFCE = Per capita final household consumption expenditures, PCEC = Per capita energy consumption, PCR = Per capita remittance, PCCI = Per capita capital investment, PFDI = Per capita foreign direct investment, PGR = Population growth rate, INR = Inflation rate, ECT = Error-correcting term. [+] and [−] present positive and negative partial sum of squares, respectively.

also bolstering economic activity and infrastructural developments, generating employment opportunities, and positively impacting living standards [72].

Moreover, the population growth rate and the inflation rate are found to have adverse asymmetric effects on per capita GDP, consistent with the findings of Ngoc [73], who also found that the inflation rate has an asymmetrically negative impact on economic growth in Vietnam, a post-conflict environment. Population growth also postulates adverse effects on economic growth of diverting resources from productivity-enhancing technologies and industries toward economic growth, which are assumed to have lower rates of return. Since substantial costs are required to offer certain services to citizens, rapid growth in the population posits a discouraging task for a war-torn society [74] and [75]. To provide more insights, the findings suggest delving into the short-run and long-run effects. Table 7 reports the results of the short- and long-run asymmetric ARDL model using Eq (9).

The results shown in Table 7 are consistent with the theoretical concept of the effects of war on the economy. The findings indicate that the war had a significant impact on the Afghan economy in both the short and long run. It shows that positive partial sum changes in the per

capita cost of war, say, an injection of one million dollars into the per capita cost of war, decreases per capita GDP by $10.099 and $15.149 in the short and long run, respectively. It further reveals that negative partial sum changes in the per capita cost of war have positive effects on the economy. It implies that lowering the per capita cost of war by a million dollars raises per capita GDP by $23.436 and $56.438 in the short and long run, respectively. The results are consistent with most economic models predicting that military spending on armed conflicts diverts economic resources from productive uses such as consumption and investment, ultimately reducing economic growth and employment opportunities. Again, the empirical findings are consistent with those presented by Gupta et al. [76], Gaibulloev and Sandler [77], Gates et al. [33], and Ezeoha and Ugwu [78] on five key significant impacts of war on the economy, among all others. First, it is found that war is linked to lower economic growth and higher inflation rates, impeding the overall economic performance. This effect causes a double impact on the economy, such as rises in general prices of goods and a substantial loss in per capita income. Second, it has a negative impact on investment and taxation (see also, [79]), suppressing the economy by limiting the public revenue conduits and distortion in the process of the formal economy. Third, it leads to increased government spending on defense. Fourth, the war brings changes in the composition of government spending, causing a significant negative impact on growth due to substantial reallocation. This also implies that smaller investment shares in infrastructure, health, education, and technology and increased government spending on war cause the higher crowding-in of government spending to be the dominant influence.

For the control predictors, the results indicate that except for population growth, which does not pose short-run effects on the per capita GDP, all other predictors have a significant negative impact on the economy both in the short and long run. Specifically, the partial sum of squares of per capita final household consumption expenditure increases per capita GDP, while its negative partial sum of squares has an adverse effect in the short and long run. The results concur with the findings of Al Gahtani et al. [80] who also highlighted the rising importance of household consumption expenditure to growth-targeting strategies in developing economies. Moreover, the positive (negative) partial sum of squares of per capita energy consumption is found to decrease (increase) per capita GDP by $10.937 ($2.142) in the short-run and by $3.327 ($10.942) in the long-run, respectively. The results are consistent with those of Zhang et al. [81], Dar et al. [82], Sek [83], and Osobajo et al. [84], who statistically found the negative impact of energy consumption on growth for China and a panel of 77 economies.

Furthermore, the findings reveal that the positive partial sum of squares of per capita remittance adversely affects the economy while its negative partial sum of squares increases the per capita GDP in the short and long runs. The results are contrary to the expected sign of the coefficients. The positive effect of remittances on growth can be empirically achieved in secured economies (see, for instance, Adjei et al. [85]), while it may respond adversely to the effects of remittances in war-torn societies, as the results presented by Ahmed [86] for some African countries. The findings show that population growth's partial sum of squares reduces per capita GDP and that its negative partial sum of squares increases growth only in the long run. The findings contrast with Thornton's [87] findings and are consistent with Peterson's [88] study, in which the former found no evidence of a long-run relationship between the economy and population growth rate and the latter found evidence that population growth reduces economic growth. Moreover, the results presented in Table 7 demonstrate that the positive (negative) partial changes in per capita capital investment and per capita FDI increase (decrease) per capita GDP both in the short and long runs.

This implies that FDI promotes technology spillovers, human capital development, and a more competitive business environment. All of these factors foster economic growth, which is

critical for alleviating poverty and raising living standards. The results are consistent with the findings of Hayat [89], Muse and Mohd [90], and An and Yeh [91], who also provided statistical evidence on the significant effects of FDI and capital investments on economic growth. Lastly, the results confirm the asymmetric effects of the inflation rate on the economy both in the short and long runs. It shows that in the short-run, a positive partial change in the inflation rate causes per capita GDP to reduce by $10.851, while its negative partial change increases per capita GDP by $5.773, noting that the long-run asymmetric effects are higher than the short-run.

The results of the asymmetric short and long-run effects of war and control predictors on per capita GDP are in line with the theoretical expectations, while they also indicate some important highlights that require further insights into the asymmetric effects of the predictors on growth. In this faith, the present study computes the dynamic multipliers, using Eq (10) to test the response of per capita GDP towards the temporal dynamics behavior of the per capita cost of war and other augmented predictors, considering the backgrounds invented by the short-run dynamics and the initial disequilibrium due to the shocks to growth confirmed by the results shown in Tables 4 and 5. The results of the dynamic multipliers are depicted in Fig 1 and demonstrate that the cumulative effect of the per capita cost of war on per capita GDP shows that the positive asymmetric shock from the per capita cost of war causes the per capita GDP to increase, while its negative asymmetric shock reduces the per capita GDP. The results are in line with the argument of Luca [92] on the wider spectrum of effects of war on the economy, but closely confirm the latest findings of Getzner and Moroz [93] relevant to the swift response of the economy to war in Ukraine. The results of the cumulative effect of per capita final household consumption expenditure indicate a counter-example, showing that growth does not significantly respond to the positive asymmetric shock of per capita final household consumption expenditure, while it positively responds to the negative shock. Moreover, the per capita GDP slightly responds to both the positive and negative asymmetric shocks from the per capita remittance, while it strongly responds to the per capita FDI. It reveals that the positive asymmetric shock from per capita FDI causes the growth to increase, whereas the growth decreases by an asymmetric negative shock from per capita FDI. The finding concurs with those of Asunka et al. [94] and Hobbs et al. [95], who also found the significant response of economic growth to FDI asymmetries. Despite the fact that the response of growth remains null to the asymmetric positive partial shock from population growth, it significantly reacts to the positive asymmetric shock from per capita capital investment. On the other hand, the significance of the growth response to the negative shock of population growth is short-term, while it becomes null and shows a decline in the long-run.

The results are statistically robust and valid, upon which the present study draws conclusions. The rare part of Table 7 reports the relevant diagnostic checks and demonstrates that the estimation does not suffer from serial correlation and heteroskedasticity problems, while it also indicates that the model is appropriately fit for purpose and the residuals are normally distributed. Moreover, the stability of the coefficients and the asymmetric ARDL model is tested by the use of the CUSUM and CUSUMSQ methods shown in Fig 2, indicating that the residuals are within the 5% bound and confirming the stability of the coefficients and the model.

## 5.4 Asymmetric causality test results

This section concludes the statistical analysis and presents the asymmetric causality analysis among the predictors, using Hatemi-J's [64] method based on the modified VAR model of Toda and Yamamoto's [65] approach. The motivation to delve into the asymmetric causality nexus is based on the critical results presented in Tables 4–7 on the one hand, while on the

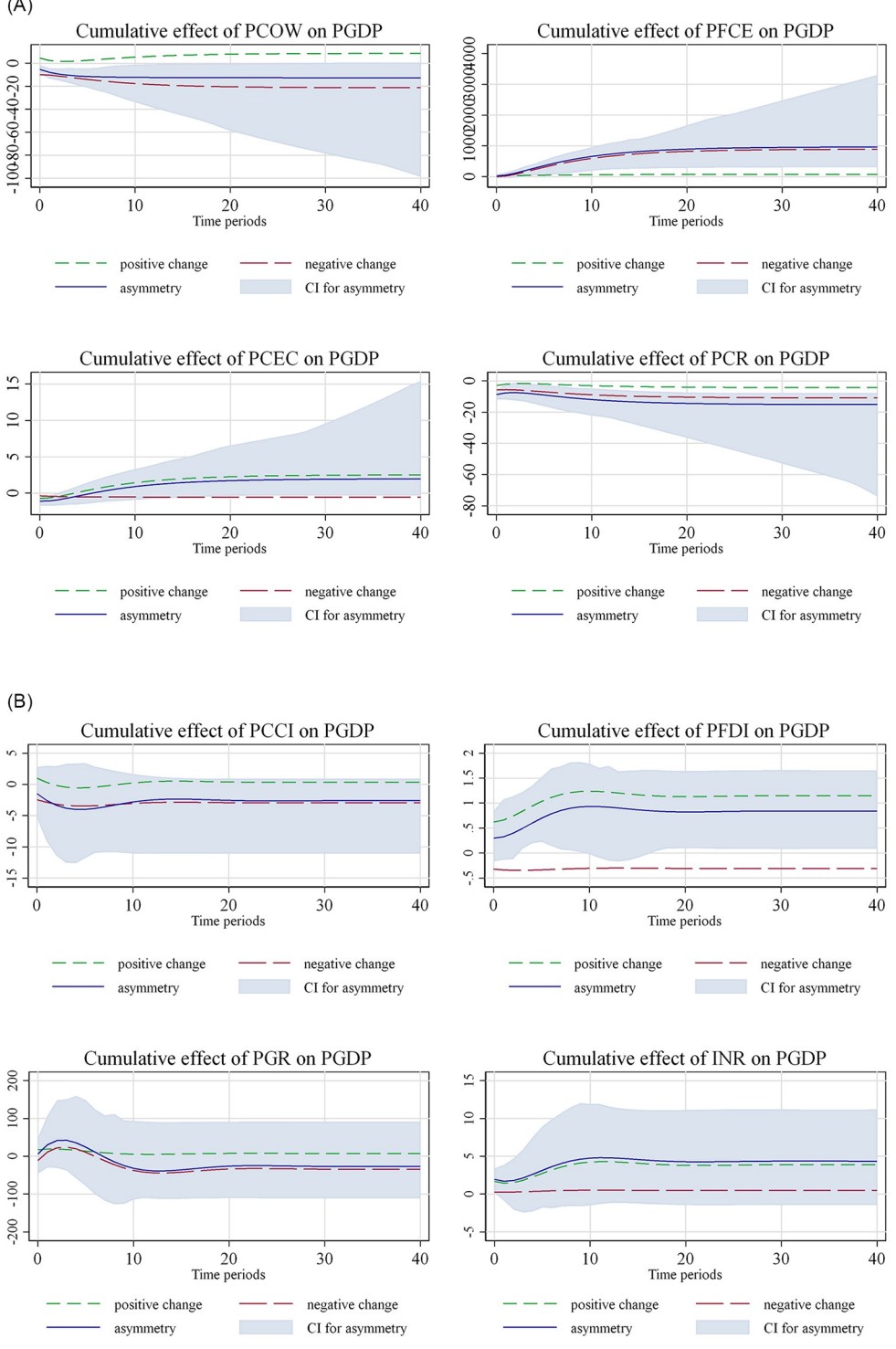

**Fig 1. Asymmetric ARDL; dynamic multipliers.** Note: 95% confidence interval bootstrap is based on 100 replications.

(A)

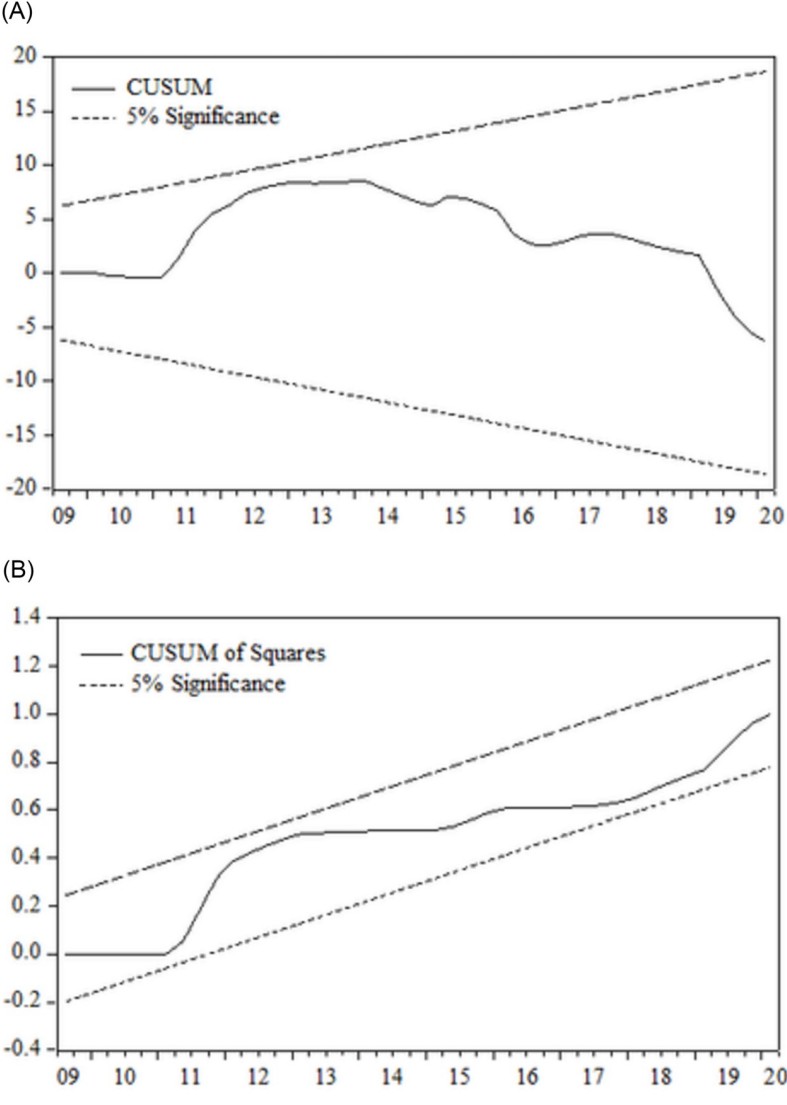

(B)

Fig 2. Asymmetric ARDL CUSUM and CUSUMSQ tests.

other hand, it is one of the benchmarks of the study to explore the war-growth interdependence. The estimation uses optimal lag length selected by the AIC, SIC, and HQIC frameworks with (dmax = 2 + 8) in the modified unrestricted vector autoregressive order and the bootstrap of 1,000 replications based on the asymptotic chi-squared distribution for the Wald test. The results are reported in Table 8 and provide interesting findings. They show that positive (negative) shocks from the per capita cost of war have a significant causal relationship with the per capita GDP, while the flip-side results confirm a significant bidirectional causality among them at 1% significant level. Furthermore, the results show that both positive and negative asymmetric shocks from per capita final household consumption expenditure, per capita capital investment, and per capita FDI have significant bidirectional causality relationships with per capita GDP and vice versa. The estimation reveals that both positive and negative asymmetric shocks of population growth are insignificant to reject the null of no asymmetric causality both for unidirectional and bidirectional nexus, while unidirectional causality is confirmed from per capita remittance and inflation rate with the per capita GDP.

**Table 8. Asymmetric causality results.**

| Causality direction | $d_{max}$ | Test statistics | Causality direction | $d_{max}$ | Test statistics | Critical values | |
|---|---|---|---|---|---|---|---|
| | | | | | | 1% | 5% |
| $PCOW_{t-i}^{+} \rightarrow PGDP_{t-i}^{+}$ | 2+8 | 18.332*** | $PCCI_{t-i}^{+} \rightarrow PGDP_{t-i}^{+}$ | 2+8 | 10.063*** | 6.781 | 3.331 |
| $PCOW_{t-i}^{-} \rightarrow PGDP_{t-i}^{-}$ | 2+8 | 33.061*** | $PCCI_{t-i}^{-} \rightarrow PGDP_{t-i}^{-}$ | 2+8 | 11.937*** | 6.781 | 3.331 |
| $PFCE_{t-i}^{+} \rightarrow PGDP_{t-i}^{+}$ | 2+8 | 15.239*** | $PFDI_{t-i}^{+} \rightarrow PGDP_{t-1}^{+}$ | 2+8 | 7.710*** | 6.781 | 3.331 |
| $PFCE_{t-i}^{-} \rightarrow PGDP_{t-i}^{-}$ | 2+8 | 12.737*** | $PFDI_{t-i}^{-} \rightarrow PGDP_{t-i}^{-}$ | 2+8 | 7.082*** | 6.781 | 3.331 |
| $PCEC_{t-i}^{+} \rightarrow PGDP_{t-i}^{+}$ | 2+8 | 14.162*** | $PGR_{t-i}^{+} \rightarrow PGDP_{t-1}^{+}$ | 2+8 | 3.011 | 6.781 | 3.331 |
| $PCEC_{t-i}^{-} \rightarrow PGDP_{t-i}^{-}$ | 2+8 | 17.917*** | $PGR_{t-i}^{-} \rightarrow PGDP_{t-i}^{-}$ | 2+8 | 2.889 | 6.781 | 3.331 |
| $PCR_{t-i}^{+} \rightarrow PGDP_{t-i}^{+}$ | 2+8 | 6.371** | $INR_{t-i}^{+} \rightarrow PGDP_{t-i}^{+}$ | 2+8 | 9.485*** | 6.781 | 3.331 |
| $PCR_{t-i}^{-} \rightarrow PGDP_{t-i}^{-}$ | 2+8 | 5.628** | $INR_{t-i}^{-} \rightarrow PGDP_{t-i}^{-}$ | 2+8 | 12.337*** | 6.781 | 3.331 |
| **Reveres causality** | | | | | | | |
| $PGDP_{t-i}^{+} \rightarrow PCOW_{t-i}^{+}$ | 2+8 | 33.145*** | $PGDP_{t-i}^{+} \rightarrow PCCI_{t-i}^{+}$ | 2+8 | 8.536*** | 6.781 | 3.331 |
| $PGDP_{t-i}^{-} \rightarrow PCOW_{t-i}^{-}$ | 2+8 | 26.044*** | $PGDP_{t-i}^{-} \rightarrow PCCI_{t-i}^{-}$ | 2+8 | 8.099*** | 6.781 | 3.331 |
| $PGDP_{t-i}^{+} \rightarrow PFCE_{t-i}^{+}$ | 2+8 | 11.773*** | $PGDP_{t-1}^{+} \rightarrow PFDI_{t-i}^{+}$ | 2+8 | 10.101*** | 6.781 | 3.331 |
| $PGDP_{t-i}^{-} \rightarrow PFCE_{t-i}^{-}$ | 2+8 | 10.494*** | $PGDP_{t-i}^{-} \rightarrow PFDI_{t-i}^{-}$ | 2+8 | 13.682*** | 6.781 | 3.331 |

Notes:

***, **, and * present significance at 1%, 5%, and 10%, respectively.

PGDP = Per capita GDP, PCOW = Per capita cost of war, PFCE = Per capita final household consumption expenditures, PCEC = Per capita energy consumption, PCR = Per capita remittance, PCCI = Per capita capital investment, PFDI = Per capita foreign direct investment, PGR = Population growth rate, INR = Inflation rate.

[+] and [−] present positive and negative partial sum of squares, respectively. → presents direction of the causality.

# 6 Conclusions

The economic cost of Afghanistan's protracted war has been astounding. Life's losses cannot be overturned, and the psychological effects of the conflict will take a very long time to recover. This study hypothesized the asymmetric effects of long-run war and other well-known control predictors on economic growth in Afghanistan for the period ranging from 2002Q3–2020Q4. Employing datasets collected from WDI (World Development Indicators) and the Department of Defense Budget of the United States and using non-linear autoregressive distributed lags, dynamic multipliers, and asymmetric causality techniques to test the competing hypotheses, the initial results indicate that the predictors exhibit mixed integrating orders [I(0) and I(1) without any I(2)] and long-run asymmetric nexus. Using the asymmetric ARDL model, interesting results were highlighted by statistical evidence. It shows that a positive partial sum shock from the per capita cost of war decreases per capita GDP, while its negative partial sum shock increases it both in the short and long runs. This implies that an increase in the cost of war causes the growth to decline and vice versa. Moreover, the results indicate the partial sum of squares of per capita final household consumption expenditure increases per capita GDP, while its negative partial sum of squares has an adverse effect in the short and long run, with a counter-evidence for per capita energy consumption and per capita remittance. The findings also reveal that the positive (negative) partial changes in per capita capital investment and per capita FDI increase (decrease) per capita GDP both in the short and long runs, while the asymmetric effects of the inflation rate on the economy were found to be adverse in the runs. Employing the dynamic multipliers demonstrates that growth swiftly responds to the positive and negative asymmetric shocks from the per capita cost of war and the control predictors. Finally, an asymmetric causality model is applied to delve into the asymmetric causality nexus amid predictors. The results disclose that there is a significant bidirectional causality nexus between per capita cost of war, per capita GDP, per capita final household consumption

expenditure, per capita capital investment, and per capita FDI, while statistical evidence only confirms a unidirectional causality between per capita GDP and the remaining variables. In sum, the results conclude that war overwhelms the economy and poses severe consequences to the well-being of the Afghan nation, the leftover social and techno structures. Based on the findings, two important and yet specific policy recommendations are provided as follows:

1. As the findings show, war has a negative impact on the economy in both the short and long run. The positive asymmetries are consistent with the real outcome of the war in Afghanistan. The new chapter of international inclusion in the war of the country brought in billions of US dollars and supported the GDP to grow, though it was unrealistic growth during the past two decades. Therefore, it is recommended that policymakers switch to higher government spending on economic infrastructure development to take the pressure off long-term economic growth.

2. In relation to the negative impacts of the war on the economy, it is also found that a shock of the positive partial sum of the war decreases economic growth in the short and long runs. Thus, it is a complex challenge that calls for international lobbying to bring peace and sustain security in the country so the long-run negative impact of the war can be controlled. This recommendation includes four proposals: (i) international community engagement to lobby for and facilitate peace and security; (ii) internal political solidarity to facilitate an internal environment conducive to peace; (iii) community awareness of the impact of the war; and (iv) emphasizing the rule of law and corruption control in the country.

### 6.1 Limitations of the study

Due to the unavailability of disaggregated datasets at sectoral and state levels, the present study used aggregated datasets, which has led the study to fail in testing the sensitivity of sectoral-growth and state-based growth against the long-war in Afghanistan. Upon its availability, future studies may use disaggregated datasets to overcome this limitation.

## Author Contributions

**Writing – original draft:** Mohammad Ajmal Hameed.

**Writing – review & editing:** Mohammad Mafizur Rahman, Rasheda Khanam.

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
