## [Decision Letter · Decision Letter 0]

9 May 2022

PONE-D-22-08275Assessing the war-economic growth nexus: A case of AfghanistanPLOS ONE

Dear Dr. Mohammad Ajmal Hameed,

Thank you for submitting your manuscript to PLOS ONE. After careful consideration, we feel that it has merit but does not fully meet PLOS ONE’s publication criteria as it currently stands. Therefore, we invite you to submit a revised version of the manuscript that addresses the points raised during the review process.

Two reviewers pointed some problems to be solved in order to improve the quality of this manuscript. Please follow up the comments and revise the MS accordingly. Please submit your revised manuscript by Jun 23 2022 11:59PM. If you will need more time than this to complete your revisions, please reply to this message or contact the journal office at plosone@plos.org. Please include the following items when submitting your revised manuscript:A rebuttal letter that responds to each point raised by the academic editor and reviewer(s). You should upload this letter as a separate file labeled 'Response to Reviewers'.A marked-up copy of your manuscript that highlights changes made to the original version. You should upload this as a separate file labeled 'Revised Manuscript with Track Changes'.An unmarked version of your revised paper without tracked changes. You should upload this as a separate file labeled 'Manuscript'.

We look forward to receiving your revised manuscript.

Kind regards,

Wen-Wei Sung, M.D., Ph.D.

Academic Editor

PLOS ONE

Journal Requirements:

Reviewers' comments:

Reviewer's Responses to Questions

**Comments to the Author**

1. Is the manuscript technically sound, and do the data support the conclusions?

Reviewer #1: Yes

Reviewer #2: Yes

2. Has the statistical analysis been performed appropriately and rigorously? 

Reviewer #1: Yes

Reviewer #2: Yes

3. Have the authors made all data underlying the findings in their manuscript fully available?

Reviewer #1: No

Reviewer #2: Yes

4. Is the manuscript presented in an intelligible fashion and written in standard English?

Reviewer #1: Yes

Reviewer #2: Yes

5. Review Comments to the Author

Reviewer #1: Review of "Assessing the war-economic growth nexus: A case of Afghanistan"

Summary:

The authors empirically investigate the impact of Afghanistan's long-term war (1980 to 2020) on its economic growth. An Autoregressive Distributed Lag (ARDL) modelling approach has been applied for the data period of 2002 to 2018. The results indicate that an increase in the cost of the war increase the per capita GDP in Afghanistan for the study period, ceteris paribus. The outcomes may serve as a basis for designing policies and strategies for Afghan and international policymakers involved in Afghanistan.

Major Comments:

- Please review the writing of the paper.

- I believe that the paper is well-motivated, and the literature review is well-done.

- The ARDL econometric method is appropriate to study the hypotheses of the paper, and the method is performed well.

Minor Comments:

- Please improve the structure in Section 3.7.

Reviewer #2: Comments on “Assessing the war-economic growth nexus: A case of Afghanistan” (PONE-D-22-08275)

War is an important factor for a countries’ economy, this paper considers an Afghanistan sample to show the relationship between the cost of war and per capita gross domestic product and per cost of war. They use an ARDL method to study the long- and short-term impact of war on GDP. There are still some problems to be solved in order to improve the quality of this paper, which are:

1. Why do authors consider only the Afghanistan sample, and authors should show the generality of results of this paper?

2. Author should re-organize the literatures so that show the contribution.

3. In the empirical test part, authors should explain the reason for choosing the control variables. There would be many other factors which can damage the economy, such as global financial crisis, climate change and diseases and commodity price, does the author consider these factors? Does the author consider the variation of the number of populations?

4. Authors are encosuraged to add more robust check, such as the method of costs?

5. Some related papers would be useful, see,

[1] Zhifang He, Jiaqi Chen, Fangzhao Zhou, Guoqing Zhang, Fenghua Wen. Oil price uncertainty and the risk-return relation in stock markets: Evidence from oil-importing and oil-exporting countries. Int J Fin Econ. 2020;1–19. https://doi.org/10.1002/ijfe.2206

[2] Jie Cao, Fenghua Wen, H E Stanley, Xiong Wang. Multilayer financial networks and systemic importance: Evidence from China[J]. International Review of Financial Analysis, 2021: 101882.

[3] Xin Yang, Shigang Wen, Zhifeng Liu, Cai Li, Chuangxia Huang. Dynamic Properties of Foreign Exchange Complex Network

[4] Xian Chen, Yang Li, Jihong Xiao, Fenghua Wen. Oil shocks, competition, and corporate investment: Evidence from China[J]. Energy Economics. 2020, 89: 104819. https://doi.org/10.1016/j.eneco.2020.104819

[5] Jie Cao, H. E. Stanley, Fenghua Wen. Measuring the systemic risk in indirect financial networks[J]. The European Journal of Finance, 2021: 1-46. https://doi.org/10.1080/1351847X.2021.1958244

6. PLOS authors have the option to publish the peer review history of their article (what does this mean?). If published, this will include your full peer review and any attached files.

Reviewer #1: No

Reviewer #2: No

---

## [Author Response · Author response to Decision Letter 0]

14 Jul 2022

Rebuttal letter

July 12, 2022

Version: First revision

General

First, let us deeply thank the respected editor and the reviewers for their constructive and valuable comments that improved the quality of this study. As per the instruction, all comments are incorporated into the manuscript. The revised parts/sections in response to the reviewers' # 1 and # 2 are highlighted in yellow and green colors, respectively. The revised parts/sections correspond to the joint comments of both of the reviewers are highlighted with grey.

Response to reviewer #1:

Summary:

The authors empirically investigate the impact of Afghanistan's long-term war (1980 to 2020) on its economic growth. An Autoregressive Distributed Lag (ARDL) modelling approach has been applied for the data period of 2002 to 2018. The results indicate that an increase in the cost of the war increase the per capita GDP in Afghanistan for the study period, ceteris paribus. The outcomes may serve as a basis for designing policies and strategies for Afghan and international policymakers involved in Afghanistan.

Major Comments:

- Please review the writing of the paper.

Response: For incorporating this comment, the following actions were taken:

1. The entire paper has been deeply reviewed. Corrections are made in terms of structure, organization, expansion of the introduction section (key research questions are added, more emphasize has been made on the novelty and originality of the paper).

2. In terms of organization, the paper is reorganized to suit the comment and the journal requirement.

3. Typo errors were identified and corrected.

4. Methodology section has been substantially revised and enriched. Dynamic multiplier and asymmetric causality techniques were also added to capture the asymmetric shocks of the predictors on growth and to determine any asymmetric causality nexus amid indicators.

5. More explanation has been provided in the data section. The range of dataset is also updated from 2002–2018 to 2002–2020 in order to reflect the most updated results.

6. The separate discussion section has been removed. Appropriate explanations were provided within the results, more comparative studies have been cited to enrich the discussion of the study and proper economic intuitions were drawn and explained.

7. Conclusion section has been substantially revised and motivated.

8. Abstract has been revised and motivated.

- I believe that the paper is well-motivated, and the literature review is well-done.

Response: Still, more emphasize has been made to enrich the motivation and reflect the novelty of the paper.

- The ARDL econometric method is appropriate to study the hypotheses of the paper, and the method is performed well.

Response: Dynamic multipliers and asymmetric causality techniques were added to the methodology to capture the asymmetric effects of the shocks and asymmetric causality nexus amid predictors.

Minor Comments:

- Please improve the structure in Section 3.7.

Response: Totally revised and restructured.

Response to reviewer #2: 

War is an important factor for a countries’ economy, this paper considers an Afghanistan sample to show the relationship between the cost of war and per capita gross domestic product and per cost of war. They use an ARDL method to study the long- and short-term impact of war on GDP. There are still some problems to be solved in order to improve the quality of this paper, which are:

1. Why do authors consider only the Afghanistan sample, and authors should show the generality of results of this paper?

Response: Based on two key motives, Afghanistan is selected as the context of this study. First, it is a true representation of history's longest war and an example of multiple state failures, much of which is the outcome of ideological battles. Second, there is a scarcity of comprehensive and quantitative studies that provide empirical insights into the scope and magnitude of the effects of war on the Afghan economy. In fact, this is the novelty of this study.

2. Author should re-organize the literatures so that show the contribution.

Response: The literature review has been substantially revised both in terms of contents and organization. New studies with most recent findings have been sited in the literature and the theorization of the study has been enriched.

3. In the empirical test part, authors should explain the reason for choosing the control variables. There would be many other factors which can damage the economy, such as global financial crisis, climate change and diseases and commodity price, does the author consider these factors? Does the author consider the variation of the number of populations?

Response: We augmented more relevant variables in the model and run the PCA. In addition to the existing control variables, population growth also added as it relatively showed significant explanatory power. Please see table 2.

4. Authors are encouraged to add more robust check, such as the method of costs?

Response: As a post-estimation for the NARDL, we believe that method of cost would be irrelevant. Therefore, we avoided to estimate it and we reached to a conclusion that the existing post-estimation diagnostic checks are sufficient to document the robustness of the results.

5. Some related papers would be useful, see,

1) Zhifang He, Jiaqi Chen, Fangzhao Zhou, Guoqing Zhang, Fenghua Wen. Oil price uncertainty and the risk-return relation in stock markets: Evidence from oil-importing and oil-exporting countries. Int J Fin Econ. 2020;1–19. https://doi.org/10.1002/ijfe.2206

2) Jie Cao, Fenghua Wen, H E Stanley, Xiong Wang. Multilayer financial networks and systemic importance: Evidence from China[J]. International Review of Financial Analysis, 2021: 101882.

3) Xin Yang, Shigang Wen, Zhifeng Liu, Cai Li, Chuangxia Huang. Dynamic Properties of Foreign Exchange Complex Network

4) Xian Chen, Yang Li, Jihong Xiao, Fenghua Wen. Oil shocks, competition, and corporate investment: Evidence from China[J]. Energy Economics. 2020, 89: 104819. https://doi.org/10.1016/j.eneco.2020.104819

[5] Jie Cao, H. E. Stanley, Fenghua Wen. Measuring the systemic risk in indirect financial networks[J]. The European Journal of Finance, 2021: 1-46. https://doi.org/10.1080/1351847X.2021.1958244

Response: The above papers were studied deeply and have been very beneficial in revising the present study.

---

## [Decision Letter · Decision Letter 1]

25 Jul 2022

Assessing the asymmetric war-growth nexus: A case of Afghanistan

PONE-D-22-08275R1

Dear Dr. Mohammad Ajmal Hameed,

We’re pleased to inform you that your manuscript has been judged scientifically suitable for publication and will be formally accepted for publication once it meets all outstanding technical requirements.

Kind regards,

Wen-Wei Sung, M.D., Ph.D.

Academic Editor

PLOS ONE

Reviewers' comments:

Reviewer's Responses to Questions

**Comments to the Author**

1. If the authors have adequately addressed your comments raised in a previous round of review and you feel that this manuscript is now acceptable for publication, you may indicate that here to bypass the “Comments to the Author” section, enter your conflict of interest statement in the “Confidential to Editor” section, and submit your "Accept" recommendation.

Reviewer #2: All comments have been addressed

2. Is the manuscript technically sound, and do the data support the conclusions?

Reviewer #2: Yes

3. Has the statistical analysis been performed appropriately and rigorously? 

Reviewer #2: Yes

4. Have the authors made all data underlying the findings in their manuscript fully available?

Reviewer #2: Yes

5. Is the manuscript presented in an intelligible fashion and written in standard English?

Reviewer #2: Yes

6. Review Comments to the Author

Reviewer #2: The authors have revised the manuscript according to the reviewer's suggestion, and the quality of this paper is quite improved. I am happy to recommend the publication of this paper.

7. PLOS authors have the option to publish the peer review history of their article (what does this mean?). If published, this will include your full peer review and any attached files.

Reviewer #2: No

---

## [Editor Report · Acceptance letter]

29 Jul 2022

PONE-D-22-08275R1 

Assessing the asymmetric war-growth nexus: A case of Afghanistan 

Dear Dr. Hameed:

I'm pleased to inform you that your manuscript has been deemed suitable for publication in PLOS ONE. Congratulations! Your manuscript is now with our production department. 

Kind regards, 

on behalf of

Dr. Wen-Wei Sung 

Academic Editor

PLOS ONE